# The Opportunities of Sustainable Biomass Ashes and Poultry Manure Recycling for Granulated Fertilizers

**Ramūnas Mieldažys \*** , **Eglė Jotautienė** and **Algirdas Jasinskas**

Institute of Agricultural Engineering and Safety, Vytautas Magnus University Agriculture Academy, Studentų 15, Akademija, LT-53362 Kaunas Distr., Lithuania
\* Correspondence: ramunas.mieldazys@vdu.lt; Tel.: +37-0377-523-76

**Abstract:** A need for the disposal of poultry manure and the reduction of its impact on the environment encourages the search for cleaner and more efficient ways to utilize and recycle production waste. It is known that granulated ash and manure are the most effective alternatives for ash and manure recycling, as compared to the unprocessed product. This paper presents an investigation of ash and poultry manure recycling for granulated fertilizers. Accepted standard experimental methods were used. The physical and mechanical characteristics of the granules, elemental composition ratio, and the process of compression of the raw material mill were determined experimentally. This research shows that, when a higher ash concentration was determined, the initial bulk density was larger and the density and pressure in the granulation process increases faster. The content of ash in the raw material increased granule strength; however, when increasing the ash mass in the raw material from 25% to 50%, energy consumption increased from 6.59 kJ·kg$^{-1}$ to 17.72 kJ·kg$^{-1}$. The process of compression of the raw material mill was obtained in two stages. In the first stage of compression, the mass density varied from 3–11 kg·m$^{-3}$ and the pressure varied from 1.25–8.27 MPa. In the second pressure stage, the mass deformation was elastic and the pressure process was described by indicator functions.

**Keywords:** poultry manure; biomass ash; granules; fractional composition; physical–mechanical properties; density

---

## 1. Introduction

Framework Directive 2008/98/EC introduced the concept of a waste hierarchy into the European legislation [1]. This waste hierarchy consists of five levels, or goals, the first of which is the prevention of waste. If this is not possible, the next step is reuse, followed by recycling [2]. All products should be returned into the economic cycle, according to the circular economy principles [3]. Due to the increasing intensity of agriculture, increases of chemical, biological, and other types of waste pollution in the soil, water, and air are inescapable.

Cattle, pig, and poultry farming are the main sources of organic waste in agriculture. In particular, accumulated waste causes many problems for massive poultry farms. Manure stored in tanks for several years spreads unpleasant odors and, consequently, these farms get a lot of complaints from neighboring residents. The disposal of waste generated in the poultry industry and the further environmental impacts have encouraged the search for cleaner and more useful possibilities for disposal, such as composting, burning, and granulation. Alternative energy, additional soil nutrients (i.e., as fertilizers), building materials, and other products have been suggested as possibilities for using manure in a useful way. Litter and manure waste products from chickens contain many nutrients, such as phosphorus and potassium, and so it is appropriate to use them as organic fertilizers. Chicken manure contains all identified essential plant nutrients and its fertilizer value has been widely researched. Further

technological improvement of such fertilizers has aroused great interest in both scientific and business areas [4].

Another problem is biomass ash utilization. The implementation of the EU strategy and directives on biomass has been increasing annually in Europe. Typically, 6%–10% of the biomass of burnt wood results in ash [5]. Scientists and businesses have searched for various ways to make use of these ashes, processing them to cleaner products and, at the same time, getting rid of the ash as waste. The main purpose of the European Union's environmental policies is to take advantage of waste flow use. This is a further reason for the increasing amount of biomass ash and use perspectives for it [6]. Recently, most attention has been focused on the recycling of ashes to other products. Bio-ash use possibilities includes fertilizers [7–9], construction materials [10], and soil stabilizers [11]. Fertilizers are the most popular way to use ashes. Wood ash is very useful for growing plants, due its rich composition. The best use for ash in fertilizers is for rough soils, due to the high amount of nitrogen; also due to phosphorus and potassium deficiency [2]. Various studies of biomass ash used as fertilizer have shown beneficial effects [12–15]. The ash improves the soil structure and supplies plants with nutrients [16–18]. It has also been concluded that biomass ash is one of the most effective potassium sources and mixing it pure, as well as with organic-based manure, such as poultry manure, could meet the potassium requirements for plant-based production. With the addition of biomass ash, poultry manure becomes incredibly rich in plant macro- and micro-nutrients [19].

According to the literature, the NPK ratios of granular organic fertilizers are various. A chemical composition analysis of cattle manure granule fertilizers showed that they contain the following elements: N, 2.56%; P, 1.51%; and K, 6.0% [20]. A study found organo-mineral fertilizers to have NPK content 4-4-2, 4-3-6, and 4-4-9 [21]. In other studies, the value ranges of poultry manure were found to have pH 6.3–8.4, total nitrogen content 2.6%–5.3%, phosphorus 0.6%–3.9%, and potassium 0.7%–5.7% [22–27]. Manure and ash granules dissolve slowly and release nutrients over a long period of time, thus providing a long-standing fertilizing effect.

It is known that granulated ashes, like granulated manure, is the most effective alternative for ash and manure recycling, compared to the unprocessed product [28]. According to other studies, the usage of biomass ash is not effective without special treatment. As the particle size of ash varies, the spreading of such a material into the soil without any treatment might be difficult, as it will be distributed unevenly. The most suitable way to improve the physical–mechanical properties of ash is to granulate it. Such a granular product can be spread into the soil much more evenly. As the plasticity of biomass ash is negligible, granulation binders are required, such as clay or various organic additives [29].

Problems with dust, transportation improvement, storage, and usage problems can be solved by granulating. Biological ash is granulated with sewage sludge and lime. The addition of sewage sludge is the main reason for significantly reduced granule compression strength. The addition of lime (slaked lime) does not increase the strength of the granules. The concentrations of heavy metals (As, Cd, Cr, Cu, Pb, Ni, and Zn) in the granules are low enough, and the concentrations of nutrients (Ca, K, P) are high enough. It has been claimed that granulated fertilizers are appropriate for fertilizing forests, after the evaluation of their heavy metal and nutrient compositions [2]. Furthermore, ashes can be granulated together with manure. After blending chicken manure compost together with biochar, P is easily accessible and, so, it can be composted and used as an organic fertilizer. A study showed that, by simultaneous granulation, it is possible to increase the utilization of ash and produce valuable products [30].

Granulated ashes together with liquid manure are used in the process of production of the granules with size very close to the size of mineral fertilizers. It is possible to obtain granules of varying sizes by using a rotating drum, where the granule size depends on the ash to liquid manure ratio [6]. However, there have been many various production technologies which are not resolved till the end (ash collection, screening, drilling, and so on). Granulated ash is a significantly more cost-effective alternative for ash recycling, compared with unprocessed ash.

Granule manufacturing is an energy-intensive process. It has been determined that the power consumption of a commercial pellet mill falls within the range of 48.9–130.7 kJ·t$^{-1}$, 37–40% of which is required to compress the material. The specific energy consumption range was 59–268.2 kJ·t$^{-1}$ for pellet mills. For a given densification system, moisture content and other biomass properties (e.g., particle size distribution and biochemical composition) can significantly affect the specific energy requirements of the process [31,32]. The specific compression energy calculated for the densification of compost derived from swine solid fraction was 12.7–17.9 kJ·kg$^{-1}$ with raw material moisture content of about 10% [33].

The aim of this work is to carry out a feasibility study for the combination of agricultural waste biomass ash and poultry manure recycled into granular fertilizer by considering the technological means of raw waste material preparation, a theoretical pressure analysis, and a physical–mechanical property determination of the obtained product.

## 2. Materials and Methods

### 2.1. Object of the Research

Experimental investigations of poultry manure and biofuel ash waste preparation and conversion into granular fertilizers were carried out, in 2017, in a laboratory based at the Institute of Agricultural Engineering and Safety in Lithuania. The poultry manure used in this study was obtained from an industrial poultry farm. A total of 10 kg of manure samples were collected from different places in the poultry manure storage house. Biomass ash was collected from the industrial burner unit of a power plant of an energy company in Lithuania, which used biomass as a fuel. The biomass burned was comprised of forest residues and wastes from the wood processing industry, in the form of sawdust and chips. Proportions and mixing ratios are presented in Table 1.

**Table 1.** Ash and manure sample codes.

| No. | Sample Ratio (*wt/wt*, %) | Samples Codes |
|-----|---------------------------|---------------|
| 1 | 1 part of ash to 1 part of manure (ratio of 50/50) | 1A + 1M |
| 2 | 1 part of ash to 2 part of manure (ratio of 33.3/66.6) | 1A + 2M |
| 3 | 1 part of ash to 4 part of manure (ratio of 25/75) | 1A + 4M |
| 4 | 3 part of ash to 7 part of manure (ratio of 30/70) | 3A + 7M |
| 5 | 4 part of ash to 6 part of manure (ratio of 40/60) | 4A + 6M |

In order to produce granules, a granulator with a horizontal matrix and 6 mm diameter channels was used. After the granules cooled, their biometric parameters were evaluated, including their dimensions, volume, and density.

### 2.2. Poultry Manure and Ash Raw Material Milling Quality

In the experimental research of the poultry manure and ash wastes, a hammer mill Retsch SM 200 (Germany) was used to grind the raw material into a homogenous form.

The milling quality was determined by using a set of 200 mm diameter sieves with the following mesh sizes: 0 mm, 0.25 mm, 0.5 mm, 0.63 mm, 1 mm, 2 mm, 3.15 mm, 4 mm, and 5 mm. By sieving a 100 g mass sample with a Retsch AS 200 (Germany) sieve shaker, the following parameters were determined: height, 1 mm; sifting time, 1 min; and interval, 10 s. After sifting, the remaining mass was weighed, and the percentages of the sample fraction of particles were calculated. Each test for every sample was repeated 5 times.

Mill bulk density was determined using an empty 6 dm3 cylinder. A prepared mill of manure and ash was poured into the cylinder, up to the upper edge. The vessel with mill was weighed and the mass of the mill was calculated (EN 13040:2007).

Mill moisture content was determined in chemical laboratories, according to the standard methodology (EN 12048:1996). Samples were weighted with KERN ABJ (Germany) scales (accurate

to 0.001 g) and dried for 24 h at a temperature of 105 °C. The moisture content of each sample was calculated as a percentage. By knowing the moisture content of the mill, the quantity of dry matter of each type of organic fertilizer granule (DM) was calculated.

### 2.3. Determination of Granulation Parameters and Pressure Test

For raw material pressure test, three types of ground chicken litter manure and biomass ash samples (1A + 1M, 1A + 2M, and 1A + 4M) were prepared. A cylindrical chamber (12.2 mm in diameter) with a piston device was filled with ground poultry manure and biomass ash, up to 120 mm. The piston stroke was set to 60 mm. During the experiment, the friction on the piston and cylinder metal surface was reduced to a minimum by using graphite grease. The lateral tangential pressures on the walls were not evaluated. After the chamber was filled with ground manure and ash, the compression force dependence on the deformation of the pressed mass was determined. The resulting curves were used to calculate the characteristics of the mechanical properties of the granulated raw material. The dependence of the pressure force $F$ required for the granulation process on the deformation of the material to be pressed was determined experimentally using test machine "Instron 5960" and the command and parameter registration software "Bluehill". The tester was set up at a compression rate of 100 mm·min$^{-1}$.

A compression chamber made for the experiment was filled with manure and ash flour with mass $m$, height $H$, and bulk density $\rho_0$. When a force $F$ was applied, the compression height $x$ and mass volume $V$ changed. This process is shown in Figure 1.

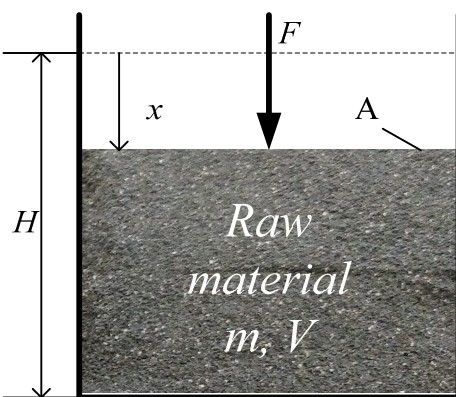

**Figure 1.** Manure and ash flour pressure piston diagram: $H$, initial height; $x$, piston stroke; $F$, force; and A, area of the pressurized surface.

The aim of this experiment was to determine the pressure necessary to obtain the desired density. It is known that the raw material pressure process of biomass is usually described by indicator functions (Equation (1)). In this case, the forces and pressures generated in the construction of the granulating equipment can be calculated according to the required product density $\rho$. There has been lot of research on the extraction of biomass from plant origins for fuel or animal feed, but no research has been carried out on the different categories of manure and ash flour or compost. In calculating the technological and technical parameters of the granulation process, it is necessary to know the required granule density $\rho$ and pressure $p$, which are described by the following dependency [34]:

$$p = C \cdot \rho^k,\tag{1}$$

where

$p$ is the required pressure, in MPa;
$C$ is a coefficient describing the mechanical properties of the granulated material;
$k$ is the coefficient of density variation; and
$\rho$ is the density required to compress a granular product, in kg·m$^{-3}$.

In our case, the required manure and ash flour density is described by the equation:

$$\rho = \frac{H}{H-x}\rho_0,\tag{2}$$

where $\rho_0$ is the initial bulk density of the raw material, in $kg \cdot m^{-3}$.

Then, the required pressure to achieve the target is found by the equation:

$$p = C\left(\frac{H}{H-x}\rho_0\right)^k,\tag{3}$$

By analyzing Equation (3), it can be seen that, when the pressed mode is set, the pressure depends on the mass height, on the initial bulk density, and on the coefficients $C$ and $k$, which characterize the physical and mechanical properties, and the values of which can be determined experimentally.

The specific compression energy, in $kJ \cdot kg^{-1}$, of manure and ash raw material can be calculated by integrating the vertical force $F$ through all displacements $x$. Compression on $\rho_0$ denotes the pressure applied to the raw manure and ash materials by the piston, as the vertical distance, in meters, through which the piston moved during the test. To calculate the compression energy $E$ as a function of density, we should use the Equation (1) with the coefficients $C$ and $k$ as determined experimentally. The resulting expression for $E$ becomes [34]:

$$E = \frac{C}{k-1}(\rho^{k-1} - \rho_0^{k-1}).\tag{4}$$

### 2.4. Granule Production

For granule production, a small-capacity 7.5 kW biomass granulator ZLSP200B (Poland) with a horizontal matrix was used, where the diameter of the matrix holes was 6 mm and its capacity was 80–120 $kg \cdot h^{-1}$. Before the ground raw material was put into the granulator, the material was mixed thoroughly (to achieve homogeneity) and moistened. After the raw material was supplied to the press chamber, the mill was moved by rollers through the matrix holes (with 6 mm diameter) and was pressed through the holes to form the granules.

### 2.5. Determination of Granule Physical–Mechanical Properties

The granule parameters of length and diameter were determined by measurement with a digital Vernier caliper LIMIT 150 mm (PRC), which met the requirements of DIN 863 (accuracy to 0.01 mm). Experimental tests were randomly selected for each granular sample, with 10 granules used to obtain the average value and error.

Granule weight was assessed by KERN ABJ (Germany) scales (accurate to 0.001 g). The weights were calculated for each type of granule, using 10 granules with the average mean error.

Granule volume was calculated using the granule size (diameter and length). After determination of granule volume and weight, the density of all investigated granule samples was calculated. The density of the samples was determined using an empty 6 $dm^3$ cylinder. The cylinder was filled up with granules, up to the upper edge, and then the cylinder's weight was measured. Using the results, the density of the material was calculated.

Granule strength tests were performed with a 5 kN capacity Instron 5965 test machine (ITW, Norwood, USA), where the load frames met the requirements of EN61236-1 (2006). The received parameters were saved to the Instron Bluehill test control software (version 3.11.1209). The cylindrical granules were horizontally placed between two anvils and a compressive force was applied to the side of the granule. The tester was run at a compression rate of 20 $mm \cdot min^{-1}$ and was stopped after the granule had fractured or broken. Experiments with the samples of granules were carried out in the horizontal and vertical planes. Each test was performed 5 times per granule type sample. The experiment results were recorded every 0.1 s, until the granule had disintegrated. Measurement

error was 0.02%. Statistical methods were used for processing the obtained data. The average values and their confidence intervals (CI) were calculated at a probability level of 0.95.

*2.6. Granule Elementary Composition Content*

The elementary composition of granules was determined at the Agrochemistry Research Centre laboratory of the Lithuanian Centre for Agrarian and Forest Sciences using a standard methodology. The pH of the experimental granules was determined according to standard ISO 10390:2005 "Soil quality. Determination of pH". The elementary composition was determined according to the standards presented in Table 2.

**Table 2.** Chemical elementary composition methods.

| Element | Standard |
|---|---|
| Nitrogen (N) | EN 13342:2000 "Characterization of sludges—Determination of Kjeldahl nitrogen" |
| Phosphorus (P) | EN 13657:2002 "Characterization of waste—Digestion for subsequent determination of aqua regia soluble portion of elements" <br> EN ISO 11885:2009 "Water quality—Determination of selected elements by inductively coupled plasma optical emission spectrometry" |
| Potassium (K) | EN 13657:2002 "Characterization of waste—Digestion for subsequent determination of aqua regia soluble portion of elements" <br> ISO 9964-3:1993 "Water quality—Determination of sodium and potassium. Determination of sodium and potassium by flame emission spectrometry" |
| Cadmium (Cd) | EN 13657:2002 "Characterization of waste—Digestion for subsequent determination of aqua regia soluble portion of elements" <br> EN ISO 15586:2003 "Water quality—Determination of trace elements using atomic absorption spectrometry with graphite furnace" |
| Zinc (Zn), Nickel (Ni), Lead (Pb), Copper (Cu) | EN 13657:2002 "Characterization of waste—Digestion for subsequent determination of aqua regia soluble portion of elements" <br> ISO 8288:2002 "Water quality—Determination of Cobalt, Nickel, Copper, Zinc, Cadmium and Lead—Flame atomic absorption spectrometric methods" |
| Chrome (Cr) | EN 13657:2002 "Characterization of waste—Digestion for subsequent determination of aqua regia soluble portion of elements" <br> ISO 9174:2003 "Water quality. Determination of chromium. Atomic absorption spectrometric methods" |

## 3. Results and Discussion

The following raw materials were used to produce granular fertilizers: manure from a poultry farm, ashes from a biomass burning enterprise, and water was used to add moisture into the mixture. The granulation process relies on capillary forces to be effective. This requires the addition of moisture throughout the granulation process, typically through the inclusion of a liquid binding agent (mostly in the form of water).

The technological scheme of manure and ash mixture fertilizers production is presented in Figure 2.

Technological operations consisted of mixing manure and ash in specific proportions, grinding, and granulation. For analysis of the physical-mechanical properties by pressure test, the first three variants (1A + 1M, 1A + 2M, and 1A + 4M) were chosen, as these variants included the maximum and minimum volumes of manure and ash.

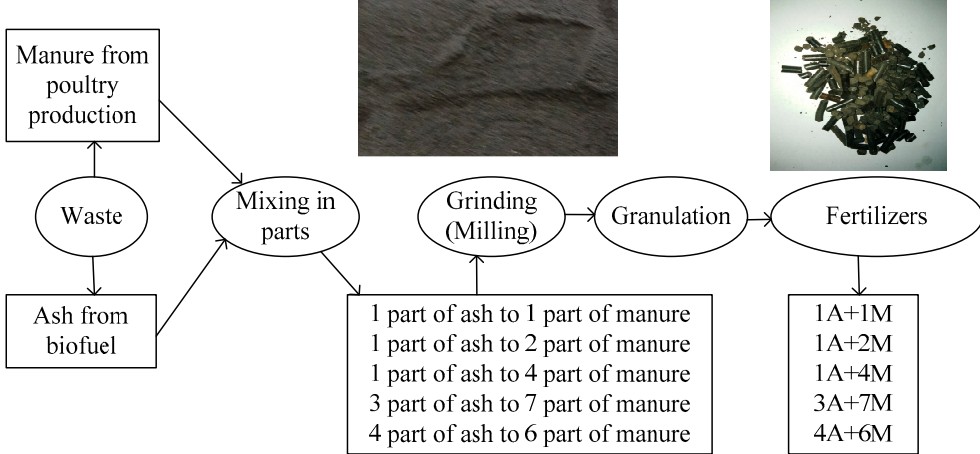

**Figure 2.** Technological scheme of manure and ash mixture fertilizers production.

### 3.1. Determination of Raw Material Mill Physical–Mechanical Properties

The physical–mechanical characteristics of moisture content and bulk density of the hammer-milled poultry manure and biomass ash waste were determined. These determined properties are presented in Table 3.

**Table 3.** The moisture content and bulk density of organic raw material.

| Raw Material Sample Name | Raw Material Moisture Content, % | Raw Material Bulk Density, kg·m$^{-3}$ |
|---|---|---|
| 1A + 1M | 16.42 ± 0.36 | 485.7 ± 0.92 |
| 1A + 2M | 26.32 ± 1.54 | 437.6 ± 3.82 |
| 1A + 4M | 30.53 ± 1.00 | 456.7 ± 2.36 |
| 3A + 7M | 27.57 ± 2.79 | 415.0 ± 0.73 |
| 4A + 6M | 24.23 ± 1.31 | 438.2 ± 12.31 |

As can be seen from Table 1, the bulk density of mill 1A + 1M was the highest at 485.7 ± 0.92 kg·m$^{-3}$ and the density of 3A + 7M was the lowest at 415.0 ± 0.73 kg·m$^{-3}$. After grinding, the moisture content of the raw material was 16.42 ± 0.36% for granules 1A + 1M, which was the lowest among the samples, due to the higher amount of dry ash. The moisture content of the other samples ranged from 24%–30%.

For sample milling, a hammer mill with a 5 mm sieve was used. Fractional composition after milling of the investigated mixture samples was determined by using sieves with holes of various diameters. Fractional composition of the prepared mill (%), depending on the sieve hole diameter (mm), is presented in Table 4.

**Table 4.** Fractional composition of mixture mill.

| Sample Code, % | Diameter Range of Sieve Holes, mm | | | | | | | | |
|---|---|---|---|---|---|---|---|---|---|
| | 0–0.25 | 0.25–0.5 | 0.5–0.63 | 0.63–1.0 | 1.0–2.0 | 2.0–3.15 | 3.15–4.0 | 4.0–5.0 | <5.0 |
| 1A + 1M | 39.77 ± 7.07 | 17.60 ± 2.95 | 8.27 ± 1.81 | 11.83 ± 0.70 | 18.33 ± 1.22 | 3.57 ± 0.74 | 0.43 ± 0.11 | 0.20 ± 0.01 | - |
| 1A + 2M | 31.20 ± 4.81 | 21.00 ± 0.32 | 12.47 ± 0.64 | 19.77 ± 1.38 | 15.37 ± 3.02 | 0.20 ± 0.18 | - | - | - |
| 1A + 4M | 16.40 ± 1.29 | 20.03 ± 1.96 | 12.37 ± 0.85 | 18.53 ± 0.46 | 21.57 ± 1.22 | 4.60 ± 1.10 | 1.37 ± 0.59 | 1.13 ± 0.11 | 4.00 ± 2.23 |
| 3A + 7M | 23.87 ± 4.44 | 18.07 ± 3.73 | 11.27 ± 1.04 | 16.43 ± 0.56 | 25.37 ± 5.57 | 4.40 ± 1.63 | 0.50 ± 0.32 | 0.10 ± 0.18 | - |
| 4A + 6M | 30.13 ± 2.27 | 17.60 ± 4.42 | 9.83 ± 0.64 | 15.03 ± 0.70 | 22.93 ± 1.33 | 3.80 ± 0.55 | 0.50 ± 0.18 | 0.17 ± 0.11 | - |

Having evaluated the fraction composition of the mills, the largest amount of sample 1A + 1M was accumulated by the 0–0.25 mm sieve (39.77 ± 7.07%). The largest amount of 1A + 2M was accumulated by the 0–0.25 mm diameter sieve (31.20 ± 4.81%). For the 1A + 4M sample, the largest amount was accumulated by the 1.0–2.0 mm diameter sieve. The 4A + 6M sample was quite similar to the 1A + 1M and 1A + 2M samples. There was no fraction accumulated by the sieve with 5 mm diameter, except for the 1A + 4M sample (4.00 ± 2.23%). The fractional composition of the raw material influences the

quality characteristics of the resultant granulated fertilizers. It can be observed, from Table 4, that manure quantity in the mixture (for example, 3A + 7M) influenced the particle size of the raw material. The particle distribution was bigger for the samples with higher manure quantities.

### 3.2. Raw Material Pressure Test

Compression force is very important for granule mechanical characteristics. During this experiment, the cylindrical chamber was filled with 3 samples (1A + 1M, 1A + 2M, and 1A + 4M) and pressed. The dependence of poultry manure and biomass ash density on compression is presented in Figure 3. According to the obtained curves (Figure 3), the changing mass density and pressure (Equation (1)) and the density change coefficient *k* were calculated. The desired compression can be selected, according to which we can obtain the corresponding granule density.

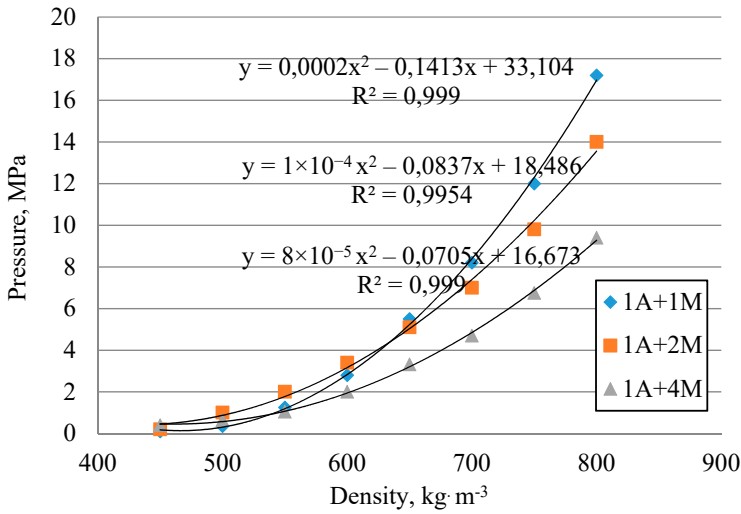

**Figure 3.** Deformation dependence on compression of the raw material.

The compression process is described by the Equation (1) and the indicator functions describing this equation. For this purpose, a function was determined experimentally. The maximum density of our laboratory equipment (cylindrical chamber) was 1032 kg·m$^{-3}$. The strength and stability of the granules obtained depended directly on the density, which should be at least 600–800 kg·m$^{-3}$.

Pressure force change, depending on material composition, is presented in Figure 4. Primary data on the compression process were derived from the experimental curve (Figure 4).

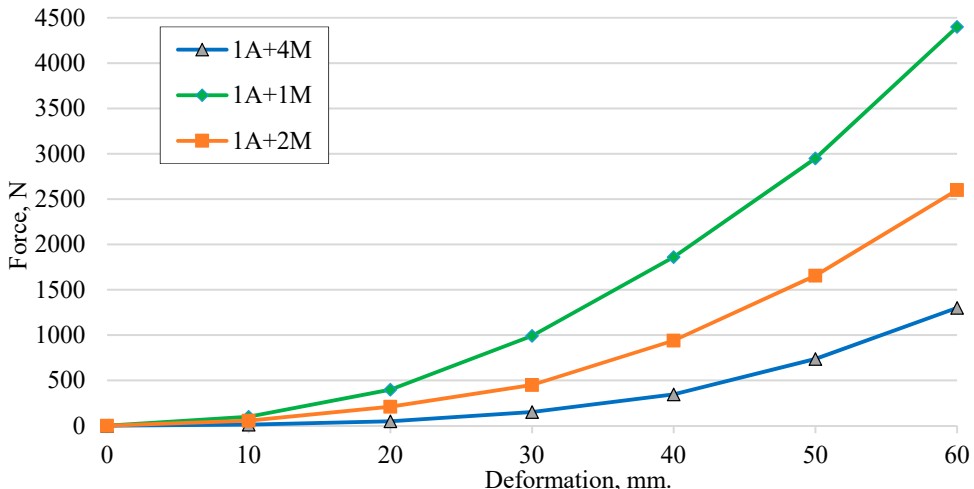

**Figure 4.** Pressure forces change depending on material composition.

Using the above-described experimental curves for calculation, we obtained the dependence of the density change on the mass deformation. Calculations were carried out throughout the pressure range, from initial density to the required density. The results are shown in Figure 5.

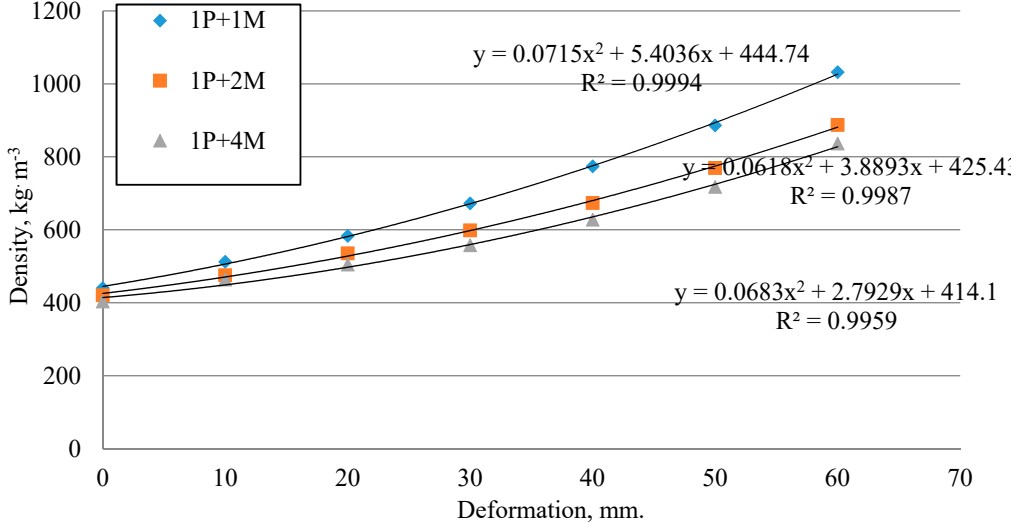

**Figure 5.** Dependence of raw material deformation on compression.

According to standard methodology, the pressure and density changes of the pressure process must be analyzed throughout pressure stages I and II from the initial density of the raw material. It is evident, from Table 5, that pressure stage I (up to 30 mm) exhibited relatively low-density changes and the pressure function is close to linear. At this stage of pressure, air is removed from the material. In stage II (from 30 mm upwards), intensive changes in density and pressure begin, which are described by indicator functions.

**Table 5.** Dependence of changes in density $\Delta\rho$ and pressure $\Delta p$ on the deformation of the compressive mass.

| Deformation, mm | | 10 | 20 | 30 | 40 | 50 | 60 |
|---|---|---|---|---|---|---|---|
| 1P + 1M | $\Delta\rho$, kg·m$^{-3}$ | 72 | 77 | 83 | 102 | 112 | 146 |
| | $\Delta p$, Mpa | 0.82 | 2.50 | 4.90 | 7.60 | 8.70 | 10.20 |
| 1P + 2M | $\Delta\rho$, kg·m$^{-3}$ | 55 | 60 | 63 | 75 | 96 | 128 |
| | $\Delta p$, Mpa | 0.46 | 1.25 | 2.00 | 4.10 | 5.96 | 7.80 |
| 1P + 4M | $\Delta\rho$, kg·m$^{-3}$ | 50 | 51 | 53 | 70 | 90 | 118 |
| | $\Delta p$, Mpa | 0.09 | 0.31 | 0.85 | 1.62 | 3.26 | 4.86 |

Table 6 shows the coefficients *C* and *k* of Equation (1), the compression pressure function, which essentially indicate the physical–mechanical properties of the material. In our case, the coefficient *C* may be assigned as the pressure dimension (MPa), and the coefficient *k* is the (dimensionless) density change coefficient at the individual deformation intervals throughout the compression process.

**Table 6.** Calculation of the coefficients *C* and *k* of Equation (1), according to the experimentally obtained pressure and density distribution curves.

| Deformation, mm | | 10 | 20 | 30 | 40 | 50 | 60 |
|---|---|---|---|---|---|---|---|
| 1P + 1M | *C*, MPa | $5.6 \times 10^{-4}$ | $6.44 \times 10^{-4}$ | $3.9 \times 10^{-4}$ | $1.33 \times 10^{-4}$ | $2.33 \times 10^{-5}$ | $3.09 \times 10^{-6}$ |
| | *k* | 1.16 | 1.34 | 1.53 | 1.76 | 2.01 | 2.35 |
| 1P + 2M | *C*, MPa | $4.35 \times 10^{-4}$ | $5.9 \times 10^{-4}$ | $4.2 \times 10^{-4}$ | $2.33 \times 10^{-4}$ | $7.2 \times 10^{-5}$ | $10.3 \times 10^{-6}$ |
| | *k* | 1.13 | 1.27 | 1.42 | 1.60 | 1.83 | 2.14 |
| 1P + 4M | *C*, MPa | $9.53 \times 10^{-5}$ | $2.93 \times 10^{-4}$ | $2.0 \times 10^{-4}$ | $1.33 \times 10^{-4}$ | $5.1 \times 10^{-5}$ | $9.39 \times 10^{-6}$ |
| | *k* | 1.12 | 1.25 | 1.38 | 1.55 | 1.78 | 2.07 |

In the final stage, the required energies $E$ (Equation (4)) were calculated. The results of the energy calculations are presented in Figure 6. It was found that higher ash concentrations require higher energy requirements; however, the results show that higher densities and stronger granules are obtained (for example, in the 1P + 1M sample case). Experimental research has shown that higher ash concentrations result in higher initial bulk density and faster density and pressure growth in the process.

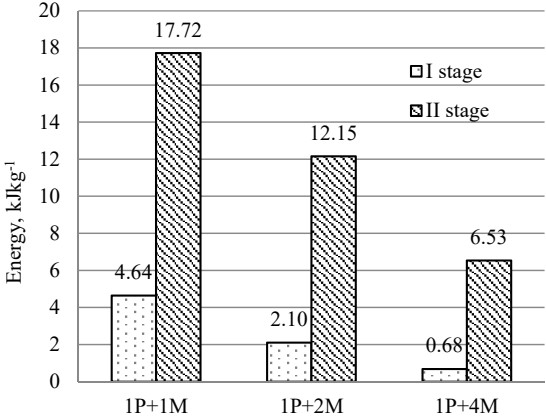

**Figure 6.** Distribution of required specific energy $E$, according to pressure stages and raw material composition.

Theoretical calculations have shown that increasing the ash concentration increases the energy consumption of the material pressing process. Increasing the ash mass in raw material from 25% to 50% resulted in an energy consumption increase, from 6.59 kJ·kg$^{-1}$ (in the case of the 1P + 4M sample) to 17.72 kJ·kg$^{-1}$ (in the case of the 1P + 1M sample). This was observed because the density of raw ash is higher than that of raw manure.

*3.3. Determination of Granule Physical–Mechanical Properties*

The average experimental granule weight was about 0.48 ± 0.3 g, and the average length was 13.9 ± 2.6 mm. Granulating organic raw material with a traditional granulator with a horizontal matrix produced a larger range of granule lengths.

It has been estimated that the mechanical properties of organic fertilizers depend on the amount of water used in irrigation and the particle size of the raw material. From the data of Table 7, the density of dry matter (DM) in 1A + 4M granules was the lowest, at 961.82 ± 59.83 kg·m$^{-3}$, and the density in 1A + 1M granules was the highest, at 1416.35 ± 99.27 kg·m$^{-3}$. Tolerable granule density is possible to achieve at a lower moisture of the raw material, but the compressive strength of the resultant granules may be considerably lower. The average density of all experimental granules after granulation was about 1452.79 ± 169.94 kg·m$^{-3}$.

**Table 7.** Densities of organic granules.

| Sample Code | Granules Density, kg·m$^{-3}$ | In Dry Materials (DM), kg·m$^{-3}$ |
|---|---|---|
| 1A + 1M | 1694.61 ± 118.70 | 1416.35 ± 99.27 |
| 1A + 2M | 1483.35 ± 182.43 | 1092.93 ± 134.41 |
| 1A + 4M | 1384.51 ± 86.13 | 961.82 ± 59.83 |
| 3A + 7M | 1386.45 ± 79.46 | 1004.21 ± 57.55 |
| 4A + 6M | 1315.01 ± 108.6 | 996.43 ± 82.29 |

The granule density obtained with the granulator was higher than that determined by the experimental method with the cylindrical chamber and Instron 5965. This is because the cylindrical chamber piston did not have the technical capabilities to achieve such densities.

### 3.4. Granule Strength Tests

Dry crush strength essentially ensures that a product will arrive at its destination as intended and can be used exactly as designed, without prematurely breaking down into finer particles. The strength test curves are shown in Figure 6. From five samples, an average inherent curve was chosen, on purpose, to show the character of the force variation in the strength test. Analyzing the deformation curves, it can be observed that the maximum crushing force was more than 338 N in the 1A + 1M sample case, in the horizontal direction (Figure 7). The other granules did not disintegrate immediately, due to their elasticity properties.

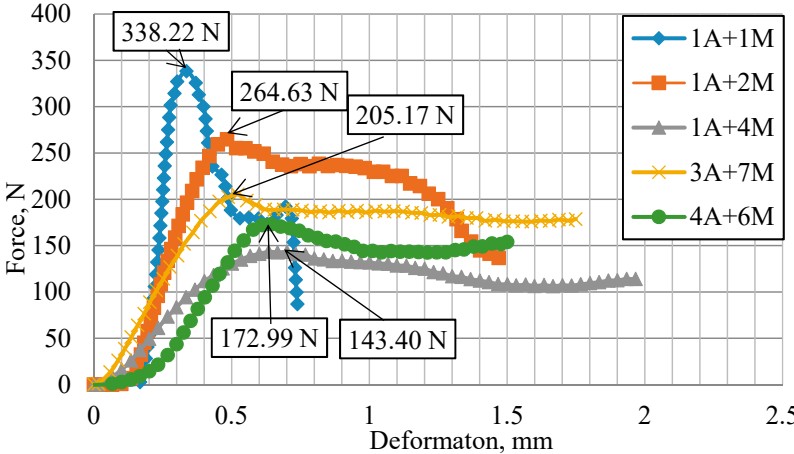

**Figure 7.** The strength test of granules in horizontal orientation.

It was determined that all experimental granules samples, under a force of 140 N in the horizontal direction, were totally disintegrated. The experimental results presented in Figure 8 show that the average strength of the granules was around 220.4 ± 75.6 N in the horizontal direction. Increasing the concentration of ash has presented stronger binding properties in the granules. The manure and ash granules (1A + 1M), with a static stability of 312.6 N (median value), were found to be the most mechanically stable. The static stability of granules ranged from 140.3 N (1A + 4M case) to 312.6 N (1A + 1M). On the other hand, the weaker granules are more suitable for dissolving fertilizers into the soil.

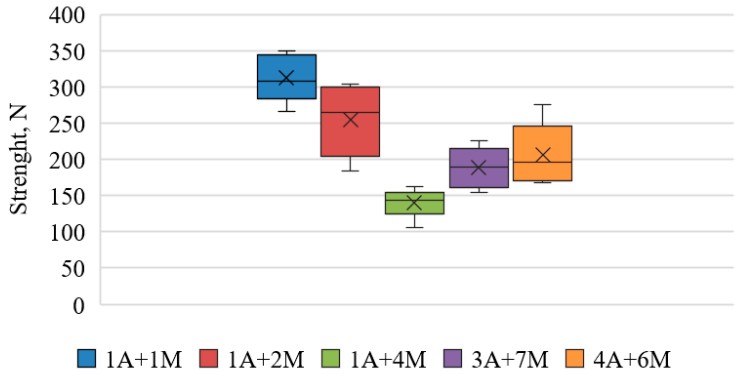

**Figure 8.** Box and Whisker plot for different granules maximum crushing load in horizontal orientation.

The same strength experiments were carried out with a vertical orientation. Analyzing the deformation curves, it can be observed that the maximum crushing force was more than 388 N in the 1A + 1M sample case, for the vertical direction (Figure 9).

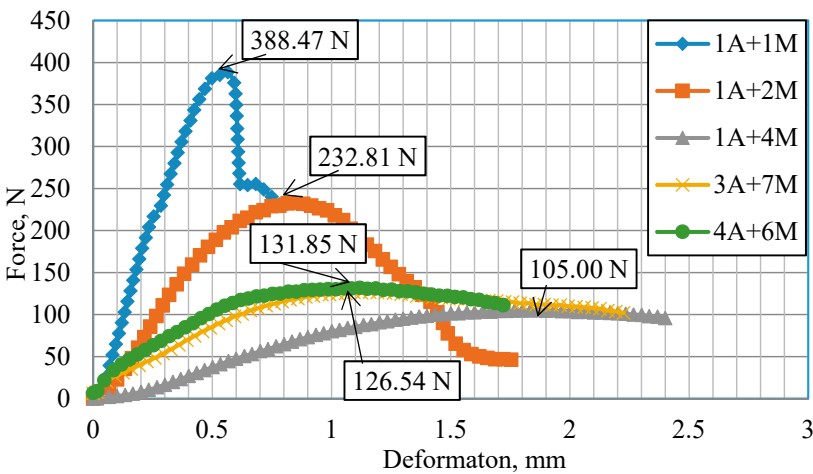

**Figure 9.** The strength test of granules in vertical orientation.

It was determined that mostly all experimental granules samples, under a force of 100 N, were totally disintegrated in the vertical direction (Figure 10). The experimental results show that the average strength of all granules was around 181.6 ± 108.7 N in the vertical direction. The average static stability of granules ranged from 101.5 N (in the 1A + 4M case) to 327.0 N (in the 1A + 1M case) in the vertical direction. Overall, all granules, except for the 1A + 1M sample, had a low variance in distribution.

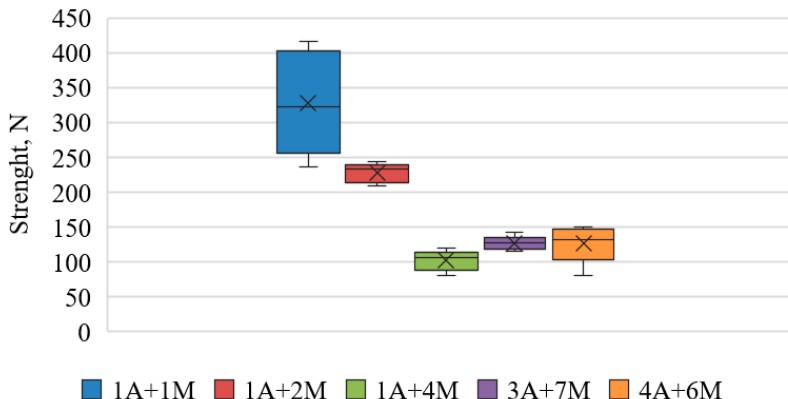

**Figure 10.** Box and whisker plot for different granules maximum crushing load in vertical orientation.

The tested granules were sufficiently resistant to operating loads, were easy to transport, and can be mechanically spread and inserted into the soil.

### 3.5. Evaluation of the Chemical Composition of Organic Granules

The experimental elemental composition ratio analysis of the granules showed Nitrogen (N) contents of 1.50%–2.27%. The N content increased as the manure content of the granules increased (e.g., the 1A + 4M case), but remained quite low (approximately 1.5%) with a relatively low content of manure (e.g., the 1A + 1M case). The phosphorus (P) content varied from 1.11%–1.29%, potassium (K) content varied from 2.12%–3.18%, and other chemical composition were small in content (see Table 8). The best NPK content was in the 1A + 4M case (NPK 2.27-1.24-2.18); however, according to the previous strength tests, the 1A + 4M sample granules were the weakest (see Figure 8).

**Table 8.** Chemical composition of tested granules.

| Test Parameters | Sample Code and Test Results | | | | |
|---|---|---|---|---|---|
| | 1A + 1M | 1A + 2M | 1A + 4M | 3A + 7M | 4A + 6M |
| pH | 10.3 | 7.9 | 8.0 | 8.8 | 9.2 |
| In dry matter: | | | | | |
| Nitrogen (N) % | 1.50 | 1.89 | 2.27 | 2.14 | 1.95 |
| Phosphorus (P) % | 1.29 | 1.16 | 1.24 | 1.16 | 1.11 |
| Potassium (K) % | 3.18 | 2.50 | 2.18 | 2.22 | 2.12 |
| Cadmium (Cd) mg·kg$^{-1}$ | 6.85 | 6.29 | 4.01 | 6.88 | 8.08 |
| Zinc (Zn) mg·kg$^{-1}$ | 5350 | 4567 | 2800 | 4100 | 5000 |
| Nickel (Ni) mg·kg$^{-1}$ | 18.6 | 21.5 | 19.5 | 18.2 | 17.8 |
| Lead (Pb) mg·kg$^{-1}$ | 288 | 217 | 142 | 206 | 264 |
| Copper (Cu) mg·kg$^{-1}$ | 59.7 | 61.0 | 60.9 | 65.6 | 64.4 |
| Chrome (Cr) mg·kg$^{-1}$ | 27.2 | 26.3 | 23.6 | 24,4 | 26.3 |

As shown in Table 8, the heavy metal contents of the granules were high. The addition of biomass ash increased the heavy metal content in all experimental samples.

Additional chemical composition experiments on the biomass ash samples were carried out, due to the high rate of heavy metal elements in the granules. The cadmium (Cd) amount was 22.8 mg·kg$^{-1}$ in ash before mixing. The lowest content of cadmium (Cd) (4.01 mg·kg$^{-1}$) was in the 1A + 4M sample, as it had the smallest quantity of ash. The decrease was greater than five times, compared to pure ash. The zinc (Zn) quantity declined from 12,233 to 2800 mg·kg$^{-1}$. The quantity of lead (Pb) declined from 719 to 142 mg·kg$^{-1}$, copper (Cu) declined from 80.4 to 60.9 mg·kg$^{-1}$, chromium (Cr) declined from 49.5 to 23.6 mg·kg$^{-1}$. The quantity of nickel (Ni) in the ash sample was 13.3 mg·kg$^{-1}$ and in the ash and manure granules, the amount of this chemical element increased to 19.5 mg·kg$^{-1}$ in the 1A + 4M sample. The smallest amount of Ni was found in the 4A + 6M sample.

Some EU member states have already detailed national legislation and guidelines for use of ash-based materials in fertilizers, including environmental requirements, such as heavy metal limits. Some countries have national regulations promoting biomass ash recycling as an alternative to landfill disposal, but harmonized EU standards regulating the use of ash-based materials are still lacking [35]. Most heavy metal contents exceeded the limit given by the preliminary requirements of the Fertilizer Regulation [36] (Table 9).

**Table 9.** Limits of heavy metals for organic fertilizers.

| Heavy Metals, mg kg$^{-1}$ | Preliminary Requirements of the Fertilizer Regulation (EC, 2016) | Maximum Concentration Allowed (U.S. EPA, 1993) |
|---|---|---|
| Cadmium (Cd) | 1.0–1.5 | 85 |
| Zinc (Zn) | 500–1500 | 7500 |
| Nickel (Ni) | 50–60 | 75 |
| Lead (Pb) | 100–120 | 420 |
| Copper (Cu) | 100–600 | 4300 |
| Chrome (Cr) | 80–100 | 3000 |

According to the preliminary Regulations of the European parliament and of the councils laying out rules on fertilizer product availability in the EU market and amending Regulations (EC) No 1069/2009 and (EC) No 1107/2009, the experimental granules do not meet the limits in the Cadmium (Cd), Lead (Pb), and Zinc (Zn) cases. However, the tested fertilizers samples met maximum concentrations allowed in the U.S. norms [37]. Most heavy metal contents exceeded the Lithuanian limits given in Rules for handling and use of wood ash. The 4A + 6M sample met all requirements for the cultivation of damaged areas, except in terms of Zinc (Zn) and Lead (Pb).

## 4. Conclusions

After evaluation of waste milling quality, poultry manure and biomass ash were milled into fractions. The biggest mill fraction was accumulated on 0.25 mm sieves (39.77% ± 7.07%) in the case of sample 1A + 1M. The largest amount of 1A + 2M was also accumulated on the 0–0.25 mm diameter sieve (31.20% ± 4.81%), while the 1A + 4M sample accumulated mostly on the 1.0–2.0 mm diameter sieve. The particle distribution was larger in samples with higher manure quantities.

The determined moisture content of the raw materials ranged from 16.42% ± 0.36% to 30.53 ± 1.00%. The bulk density of the raw material was the lowest in the 3A + 7M sample case (415.0 ± 0.73 kg·m$^{-3}$) and the bulk density of 1A + 1M was the highest (485.7 ± 0.92 kg·m$^{-3}$), which showed the best results in the granule strength experiments.

The strength and stability of the granules obtained depended directly on the density, which should be at least 600–800 kg·m$^{-3}$. The granule density obtained from the granulator was obtained from 1416.35 ± 99.27 kg·m$^{-3}$ DM (in the 1A + 1M case) to 961.82 ± 59.83 kg·m$^{-3}$ DM (in the 1A + 4M case) and, with the laboratory equipment pressure chamber, a density range of 836–1032 kg·m$^{-3}$ was obtained.

The process of compression of raw material mill from poultry manure and biomass ashes can be distinguished into two stages. Air is displaced in the first stage of compression; change in mass density was only 3–11 kg·m$^{-3}$, and the pressure varied, depending on the raw material composition, from 1.25–8.27 MPa. In stage II, the mass deformation is elastic, and the pressure process is described by the indicator functions.

Increases in ash content, from 25%–50% in the raw material granules, led to strength increases; from 140.3–312.6 N on average in the horizontal direction and from 101.5–327 N on average in the vertical direction.

Theoretical calculations have shown that increasing the ash concentration increases the energy consumption of the material pressing process. Increasing the ash mass in raw material from 25% to 50% resulted in energy consumption increases, from 6.59 kJ·kg$^{-1}$ to 17.72 kJ·kg$^{-1}$.

In conclusion, the results of this study suggest that the granulation of a manure/ash mixture using biomass granulators dedicated to forestry and other agricultural residue granulation produces granules of high density (1694.61 ± 118.70 in the 1A + 1M case). It has been theoretically proven that manure and ash mixture granulators can be used at lower pressures and energy consumptions, resulting in a lower environmental impact.

**Author Contributions:** Conceptualization, E.J. and R.M.; methodology, A.J.; investigation, R.M.; writing—original draft preparation, R.M.; writing—review and editing, E.J.; supervision, E.J.

**Funding:** This research received no external funding.

**Conflicts of Interest:** The authors declare no conflict of interest.

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
