# Peer review of "The Opportunities of Sustainable Biomass Ashes and Poultry Manure Recycling for Granulated Fertilizers"

_sustainability, doi:10.3390/su11164466_

Round 1
Reviewer 1 Report
1. The work deals with quite interesting issues regarding the use of waste from both agriculture and from combustion processes. The authors only dealt with the technical side of the production of fertilizer from ash and chicken manure. They used a typical pelleting plant to prepare the fertilizer. The work mainly concerns issues in the field of mechanics and strength of materials. It should be emphasized the importance of granulate strength for its effective use as a fertilizer, consistent with the principles of sustainable development. There is a lack of utilitarian conclusions, important from the point of view of sustainable development.
2. Explain what type of biomass was burned in the power plant? What was the elemental composition of the ash formed there? It seems that because of the high rate of heavy metal elements in the pellet, this is crucial. Larger concentrations were found in granulates from mixtures containing a greater amount of biomass ash. The abnormal content of heavy metal elements in biomass ash discredits it as a fertilizer.
3. It would be advisable to supplement with information on the energy consumption of the pelleting process, whether the results of theoretical calculations coincide with the results of pelleting.
4. It would be necessary to extend the information about the pelleting machine by the value of efficiency, and information on the pelleting parameters themselves as they affect the properties of pellets obtained.
5. How was the manure for preparing mixtures and granulation prepared?
6. How does the fractional composition affect the quality of granules? (The more manure, the greater the diversity of the fraction).
7. In line 225: is "in Figure 1", should be "in Figure 2".
8. The text in lines 366-373 should be in the introduction.
Author Response
We would like to thank for very great comments of reviewer to help us improve the manuscript and to inform you that the revised version of our manuscript with considering all comments of reviewers.
Changes were made in the manuscript by using red colour.
Point 1: It should be emphasized the importance of granulate strength for its effective use as a fertilizer, consistent with the principles of sustainable development. There is a lack of utilitarian conclusions, important from the point of view of sustainable development.
Response 1: In our study it has been theoretically proven that manure and ash mixture densification can be used at lower pressures and energy consumptions, resulting in a lower environmental impact. Also, the results of this study suggest that the granulation of a manure/ash mixture using biomass granulators dedicated to forestry and other agricultural residue granulation produces granules of very high density and it is inappropriate to use them for ash/manure mixture granulation. The granules strength is important mainly for fertilizers packing, storage, transporting and storage, mechanical application in soils and dissolving into the soil. Conclusion were reviewed, the last conclusion was remade from the point of view of sustainable development.
Point 2: Explain what type of biomass was burned in the power plant? What was the elemental composition of the ash formed there? It seems that because of the high rate of heavy metal elements in the pellet, this is crucial. Larger concentrations were found in granulates from mixtures containing a greater amount of biomass ash. The abnormal content of heavy metal elements in biomass ash discredits it as a fertilizer.
Response 2: There was burned biomass from forest residues, wastes from the wood processing industry in form of sawdust (class SM2). Elemental composition of ash was added in text (in 389-397 line). We agree that abnormal content of heavy metal elements discredits it as a fertilizer, that’s why in future researches we will pay attention on it. In our study the main point was to determine possibilities of ash/manure mixtures granulation form technical point of view.
Point 3: It would be advisable to supplement with information on the energy consumption of the pelleting process, whether the results of theoretical calculations coincide with the results of pelleting.
Response 3: Theoretical calculations not coincide with the results of pelleting with pellet mill ZLSP200B because calculations were made only to cylindrical chamber. Purpose of the investigations was to find out how different ash/manure mixtures impact on energy consumption.
Point 4: It would be necessary to extend the information about the pelleting machine by the value of efficiency, and information on the pelleting parameters themselves as they affect the properties of pellets obtained.
Response 4: In the text there were added the information about granulator: standard biomass 7,5 kW granulator ZLSP200B with horizontal matrix was used for granulation, it’s capacity was 80-120 kg.h-1. The concrete granulator produces granules of too high strength and density, so we did not go into details of the granulator.
Point 5: How was the manure for preparing mixtures and granulation prepared?
Response 5: The manure was kept outside poultry farm, it was dried naturally and after that it was mixed with biomass ash.
Point 6: How does the fractional composition affect the quality of granules? (The more manure, the greater the diversity of the fraction).
Response 6: The finer the fractional composition, under the same conditions, results in higher strength and density granules. Also we can see that manure quantity in mixture (in example 3A+7M) influences particle size and that’s why there is bigger particle distribution in all diameters.
Point 7: In line 225: is "in Figure 1", should be "in Figure 2".
Response 7: It was changed to "in Figure 2".
Point 8: The text in lines 366-373 should be in the introduction.
Response 8: Text part was moved to Introduction section.

Reviewer 2 Report
Dear authors, your paper is almost interesting.
Introduction and methodology description are well written.
abstract and References quite complete.
Several mistakes (for instance fig. 2 "granullation" or verb in line 225, etc.). It is necessary a robust revision of English language
I suggest You to review conclusions
First of all, I think you have to add a "results" section.
The first conclusion sentences (lines from 395 to 407) have to be organized as "results" of the tests.
Conclusion have to contain purposes and suggestions, according with introduction, abstract and title of the paper as well. You have to conclude, thanks to your experimental part, in which way biomass ashes and poultry manure can be recycled, which advantages in terms of sustainability and regulation accordance
Author Response
We would like to thank for very great comments of reviewer to help us improve the manuscript and to inform you that the revised version of our manuscript with considering all comments of reviewers.
Changes were made in the manuscript by using red colour.
Point 1: Several mistakes (for instance fig. 2 "granullation" or verb in line 225, etc.). It is necessary a robust revision of English language
Response 1: Mistakes were corrected (Figure 2 and verb “is” in line 235). English language was reviewed by MPDI English Editing service.
Point 2: I suggest You to review conclusions
Response 2: Conclusion were reviewed, the last conclusion was remade.
Point 3: First of all, I think you have to add a "results" section.
Response 3: Results and Discussion sections were combined from beginning and in this stage it is very complicate to separate it.
Point 4: The first conclusion sentences (lines from 395 to 407) have to be organized as "results" of the tests.
Response 4: The first conclusion was supplemented.
Point 5: Conclusion has to contain purposes and suggestions, according with introduction, abstract and title of the paper as well. You have to conclude, thanks to your experimental part, in which way biomass ashes and poultry manure can be recycled, which advantages in terms of sustainability and regulation accordance
Response 5: The last conclusion was remade. We are proposing to use those materials like a fertilizers, but in granular form. There is no regulation acts for such products, it is only European Commission Circular economy documents package and proposals, also preliminary Regulations of the European parliament and of the councils laying out rules on fertilizer product availability in the EU market and amending Regulations (EC) No 1069/2009 and (EC) No 1107/2009.

Reviewer 3 Report
The aim of this work was to carry out a feasibility study for agricultural waste biomass ash and poultry manure recycling into granular fertilizer applying technological means of waste raw material preparation, theoretical pressure analysis and physical–mechanical properties determination of the obtained product. However, the revision is still needed before the acceptance of this manuscript in Sustainability. 1. In Abstract section, some items must be rewritten and simplified. 2. Table 1. “ratio of 1:1”, please indicate the unit of ratio. 3. Table 5. “2.00” and “4.1”; Table 7. “1694.61” and “118.7”. Please keep the decimal point consistent as the scientific reports. 4. Why options 3A+7M and 4A+6M were used for chemical research? This should be added. 5. Lines 237-238. “–485.7±0.92” and “–415.0±0.73”. Some references are needed. Please delete “–”. 6. Figs. 2, 7 and 9 are not very clear. Please improve resolution of the Figs, especially the numbers and words. 7. Lines 303-304. “As the ash concentration decreases from 50 to 25%, the energy demand decreases from 2.7 to 6.8 times.”. The authors should explain this result. 8. Line 364. “2.18%” should be “2.12%” based on Table 8. 9. Lines 365-366. “but according to previous experiments it was weakest granules kind in granules strength tests.”. Some references are needed. 10. The authors reported a lot of data in “Result” and “Discussion” sections. However, discussion should also be addressed to compare with the studies in the literatures. 11. At last, the level of English throughout this manuscript should be improved.
Author Response
We would like to thank for very great comments of reviewer to help us improve the manuscript and to inform you that the revised version of our manuscript with considering all comments of reviewers.
Changes were made in the manuscript by using red colour.
Point 1: In Abstract section, some items must be rewritten and simplified.
Response 1: Abstract section was rewritten.
Point 2: Table 1. “ratio of 1:1”, please indicate the unit of ratio.
Response 2: “ratio of 1:1” was changed to “ratio of 50/50”.
Point 3: Table 5. “2.00” and “4.1”; Table 7. “1694.61” and “118.7”. Please keep the decimal point consistent as the scientific reports.
Response 3: Mistake was fixed.
Point 4: Why options 3A+7M and 4A+6M were used for chemical research? This should be added.
Response 4: It was mistake. Chemical composition experiments made for all 5 samples. Part of the sentence was deleted.
Point 5: Lines 237-238. “–485.7±0.92” and “–415.0±0.73”. Some references are needed. Please delete “–”.
Response 5: “–” was deleted.
Point 6: Figs. 2, 7 and 9 are not very clear. Please improve resolution of the Figs, especially the numbers and words.
Response 6: Figures were replaced like objects. Resolution was improved.
Point 7: Lines 303-304. “As the ash concentration decreases from 50 to 25%, the energy demand decreases from 2.7 to 6.8 times.”. The authors should explain this result.
Response 7: It was mistake. Sentence was rewritten.
Point 8: Line 364. “2.18%” should be “2.12%” based on Table 8.
Response 8: It was mistake. It was changed to “2.12%”.
Point 9: Lines 365-366. “but according to previous experiments it was weakest granules kind in granules strength tests.”. Some references are needed.
Response 9: We put reference to Figure 7.
Point 10: The authors reported a lot of data in “Result” and “Discussion” sections. However, discussion should also be addressed to compare with the studies in the literatures.
Response 10: It is quite difficult to compare with other studies in literature, because other authors studied different origin raw material and form of organic granules.
Point 11: At last, the level of English throughout this manuscript should be improved.
Response 11: English language was reviewed by MPDI English Editing service.

Reviewer 4 Report
The manuscript ``The opportunities of sustainable biomass ashes and poultry manure recycling for granulated fertilizers`` was well-preparation and -written. However, I have a few comments for further improvement as follows.
1. Line 10 to 18 in Abstract and Line 39-48 in Introduction are overlapping. Abstract is the summary of your work and therefore it should summarize the most important points in your paper.
2. For ash and manure different mixing ratios, what is the unit (v/v or wt/wt)?
3. Raw materials moisture content in line 237-240 was largely depend on moisture content of manure rather than that of ash. However, there is no information on such data for each material used in this study. Therefore, Authors need to show not only moisture content of these materials but also other physio-chemical properties of ash and manure in the materials and methods section. What is the type of biomass ash?
4. Different ratios of ash and manure resulted different granule strength, energy demand, nutrient and heavy metal contents but it is difficult to understand whether these differences are significant or not without using proper statistics. Therefore, it is recommended to conduct statistical analysis to see the significant effect of different mixing ratios.
5. In some figures, authors need to explain type of error bars in figure legend (Eg. Fig. 7 and 9) and others need to add error bars.
Author Response
We would like to thank for very great comments of reviewer to help us improve the manuscript and to inform you that the revised version of our manuscript with considering all comments of reviewers.
Changes were made in the manuscript by using red colour.
Point 1: Line 10 to 18 in Abstract and Line 39-48 in Introduction are overlapping. Abstract is the summary of your work and therefore it should summarize the most important points in your paper.
Response 1: Abstract was rewritten, so there is no overlapping now.
Point 2: For ash and manure different mixing ratios, what is the unit (v/v or wt/wt)?.
Response 2: The unit is wt/wt - weight-weight (percentage).
Point 3: Raw materials moisture content in line 237-240 was largely depend on moisture content of manure rather than that of ash. However, there is no information on such data for each material used in this study. Therefore, Authors need to show not only moisture content of these materials but also other physio-chemical properties of ash and manure in the materials and methods section. What is the type of biomass ash?
Response 3:
We regret that we did not determine the properties of the starting materials of ash and manure. In this stage is impossible to do that. We were concerned about the conditions for good granulation, so we only determinate the material mixture moisture content and other physical properties (bulk density, fractional composition etc.). Due to the good granulation conditions, it was important to prepare the right moisture for the mixture and the moisture of the ingredients is not relevant in our case. By the way other physical-chemical properties (density, fractional composition, chemical composition) do not significant affect the granulation conditions. Also elemental composition of pure ash was added in text (in 389-397 line).
There was burned biomass from forest residues, wastes from the wood processing industry in form of sawdust (class SM2).
Point 4: Different ratios of ash and manure resulted different granule strength, energy demand, nutrient and heavy metal contents but it is difficult to understand whether these differences are significant or not without using proper statistics. Therefore, it is recommended to conduct statistical analysis to see the significant effect of different mixing ratios.
Response 4:
We agree that the data should be subjected to better statistical analysis. For granule strength statistically evaluated only by the average values and their confidence intervals (CI) calculation at a probability level of 0.95. The differences of samples are shown in results of Fig. 7 and 9. Granule strength is mostly important for granules suitable for packing, storage, transport and application in soils by traditional techniques and technologies. The energy consumption evaluation was carried out only on a theoretical level to compare the influence of additives of ash and manure and no statistical analysis was used in the calculations. Chemical analysis has been carried out for preliminary environmental assessment.
Point 5: In some figures, authors need to explain type of error bars in figure legend (Eg. Fig. 7 and 9) and others need to add error bars.
Response 5:
In Figure 7 and 9 the average values and their confidence intervals (CI) were calculated at a probability level of 0.95. Others figures there was not experimental test but theoretical (Fig. 2, 3, 4, 5). In Figures 6 and 8 an average inherent curve is presented, on purpose, to show the character of the force variation in the strength test.

Round 2
Reviewer 3 Report
Based on the above changes, this paper can be published.